# Unraveling the Structural Variations of Early-Stage Mycosis Fungoides—CD3 Based Purification and Third Generation Sequencing as Novel Tools for the Genomic Landscape in CTCL

**DOI:** 10.3390/cancers14184466

**Published:** 2022-09-14

**Authors:** Carsten Hain, Rudolf Stadler, Jörn Kalinowski

**Affiliations:** 1Center for Biotechnology (CeBiTec), Bielefeld University, 33615 Bielefeld, Germany; 2University Clinic for Dermatology, Johannes Wesling Medical Centre, UKRUB, University of Bochum, 32429 Minden, Germany

**Keywords:** cutaneous T-cell lymphoma, mycosis fungoides, enrichment, sequencing, nanopore, copy-number variation, structural variation

## Abstract

**Simple Summary:**

Mycosis fungoides is the most common cutaneous T-cell lymphoma, but knowledge of the genetic alterations, particularly of the early stages, is limited. A major problem is that biopsies from early stages contain few tumor cells and many “healthy” skin cells, making accurate analysis of tumor cells difficult. Here, we demonstrate a workflow to enrich tumor cells and thereby obtain better results for mutation detection, especially for deletions and amplifications. For the same sample, we also demonstrate the advantages of long-read sequencing for a more comprehensive elucidation of genetic alterations in early stages of mycosis fungoides.

**Abstract:**

Mycosis fungoides (MF) is the most common cutaneous T-cell lymphoma (CTCL). At present, knowledge of genetic changes in early-stage MF is insufficient. Additionally, low tumor cell fraction renders calling of copy-number variations as the predominant mutations in MF challenging, thereby impeding further investigations. We show that enrichment of T cells from a biopsy of a stage I MF patient greatly increases tumor fraction. This improvement enables accurate calling of recurrent MF copy-number variants such as *ARID1A* and *CDKN2A* deletion and *STAT5* amplification, undetected in the unprocessed biopsy. Furthermore, we demonstrate that application of long-read nanopore sequencing is especially useful for the structural variant rich CTCL. We detect the structural variants underlying recurrent MF copy-number variants and show phasing of multiple breakpoints into complex structural variant haplotypes. Additionally, we record multiple occurrences of templated insertion structural variants in this sample. Taken together, this study suggests a workflow to make the early stages of MF accessible for genetic analysis, and indicates long-read sequencing as a major tool for genetic analysis for MF.

## 1. Introduction

Mycosis fungoides (MF) is the most common form of cutaneous T-cell lymphoma (CTCL) and manifests as inflammatory lesions on the skin [1,2]. Early-stage MF often shows fixed skin lesions, called patches. In advanced stages of the disease, partly concurrent development of more infiltrated plaques or tumors are observed [1,3]. Furthermore, involvement of lymph nodes or visceral organs is possible [4,5,6]. Disease progression is patient-specific, with 20–30% progressing into an aggressive advanced stage disease with life expectancies between 1.5 and 4 years. The rest remains in an indolent early-stage disease with normal life expectancy [4,7,8]. A prognosis about the course of the disease is possible by clinical and histological features via the “Cutaneous Lymphoma International Prognostic index” (CLIPi) score [9,10] or via tumor clone frequency by T-cell receptor sequencing [8]. This prognosis is crucial as indolent and aggressive forms of MF require different forms of therapy: ranging in its extremes from a watch-and-wait strategy for indolent disease, to systemic therapy in the advanced form [3,11,12,13] with new targeted therapies such as brentuximab vedotin [14] or mogamulizumab [15].

Currently, a prognosis linking somatic mutations to disease progression and potential success of treatments, as (partly) present in other cancers [16,17,18], does not exist for MF. One reason why there is little correlation between genetics and possible prognosis of disease progression in MF is the lack of sufficient genetics data on the different stages of MF. While CTCL as a whole is a disease with a large genomic data source and more than 250 sequenced samples [19,20,21,22,23,24,25,26,27,28,29,30], MF datasets are a minority, accounting for only one fifth of all samples. Moreover, MF samples mostly stem from tumor-stage patients, often with a long history of different treatments. This makes it difficult to distinguish between driver mutations that initially determine disease progression, and secondary mutations that have less impact on early tumor development.

One major obstacle in analyzing early-stage MF skin lesions is the general low number of malignant cells in skin biopsies. In MF, atypical cells are found in variable numbers dependent on the disease stage [31]. During early stage, malignant cells are in the minority [32,33] and the fraction of malignant cells increases in well-established lesions in advanced stages [34]. In cases of Sézary syndrome (SS), the most often sequenced form of CTCL, tumor cells are enriched to generally high purities, as CD4+ cells from peripheral blood mononuclear cells [21,23,25,26,27] or CD3+CD26-CD14-CD8-CD19- cells from blood [20]. For MF, either whole biopsies [19,22], or microdissection of tumor cells after HE- [35] or CD3-staining [8] are used.

Most studies on CTCL genomics were done using whole-exome sequencing (WXS). This technique enables cost-effective high sequencing depth for coding sequences and shows high-sensitivity for single nucleotide variants (SNV) with limit-of-detection around 5% [36,37,38]. In contrast, WXS shows poorer sensitivity for the detection of copy-number variants (CNV) [39,40,41,42], a problem that is magnified in cancer samples with increased heterogeneity and decreasing tumor fraction [43]. As CNVs are very common and important in MF [19,22,28], careful attention on CNV calling performance in MF samples is necessary. Another disadvantage of WXS is that it only finds amplifications and deletions and not the causative structural variants (SV) leading to these CNVs. Thus, for a complete detection of SV, whole-genome sequencing (WGS) is needed.

Recent improvements in third generation sequencing makes these methods especially interesting for SV calling. Long-read lengths enable high-confidence mapping even in low complexity or repeating regions [44,45]. They allow phasing of multiple SV breakpoints into one haplotype [46] and enable the direct identification of complex SVs [47,48]. Current problems such as higher per-base error-rate are steadily reduced by new (bio)chemistry and bioinformatic tools [49].

## 2. Materials and Methods

### 2.1. Access to Restricted Data and Patient Consent

For the analysis of tumor fraction in CTCL, WGS, and WXS data of 76 SS patients from 3 studies [20,23,25], 11 MF patients from 2 studies [22,23], and 9 samples from this work were aggregated. All original studies stated that all patients gave written informed consent. The data from Choi et al. 2015 [20] and McGirt et al. 2015 [22] was publicly available on SRA. The data from Ungewickell et al. 2015 [23] and Wang et al. 2015 [25] is hosted on dbGaP with consent groups of Health/Medical/Biomedical or General Research Use, respectively. We applied successfully for both controlled datasets, clearly stating our intentions of combining multiple cohorts of patients from different studies.

### 2.2. Data Accession

Access to the controlled datasets phs000913 [23] and phs000725 [25] was requested via dbGaP. Paired-end FASTQ data from these datasets as well as the SRA datasets SRP058948 [20] and SRP059214 [22] was downloaded with the SRA toolkit 2.8.2. For each individual, WGS or WXS data for a tumor and a normal sample was downloaded. Sample SRR2046920 was an exception, since only tumor data were available.

### 2.3. Sample Collection

This study’s subject is a 79-year-old male (at the time of sample collection) with IB stage MF (T1b N0 M0 B1) of the CD4 phenotype. Written consent was obtained from the patient. A spindle biopsy from a skin lesion was taken, from which 8 × 5 mm punch biopsies were obtained and used for further analysis.

### 2.4. Tissue Disruption, Enrichment of CD3+ and CD4+ Cells, and DNA Isolation

In two aliquots, four 5 mm punch biopsies were disrupted into single cells using the human Whole Skin Dissociation Kit (Miltenyi Biotec, Bergisch Gladbach, Germany) according to the manufacturer’s protocol, with a 3 h incubation at 37 °C in each of two aliquots. Enzyme P was not added to the aliquot intended for CD4+ isolation. A small aliquot, suitable for DNA isolation, was taken from the disrupted tissue with Enzyme P added. From the remaining material, CD3+ or CD4+ cells were captured using human CD3 or CD4 MicroBeads (Miltenyi Biotec, Bergisch Gladbach, Germany) on MS Columns (Miltenyi Biotec, Bergisch Gladbach, Germany). From single cells, as well as CD3+ or CD4+ enriched cells, DNA was isolated using the DNeasy Blood & Tissue Kit (Qiagen, Hilden, Germany).

### 2.5. Whole-Exome Sequencing

For whole-exome sequencing, 150 ng of sample DNA was sheared to 300 bp length using Covaris microTUBEs (Covaris, Woburn, MA, USA). Library construction was carried out with the KAPA HyperPrep Kit (Roche, Basel, Switzerland) with IDT xGEN^®^ Dual Index UMI Adapters (IDT, Coralville, IA, USA) and four PCR cycles. Afterwards, exon capturing using the IDT xGEN^®^ Exom Research Panel v1.0 (IDT, Coralville, IA, USA) was carried out according to manufacturer’s protocol. The library was size-selected to 350–560 bp using the BluePippin device (Sage Science, Beverly, MA, USA), quantified using Qubit dsDNA HS (ThermoFisher, Waltham, MA, USA), evaluated via capillary electrophoresis using Bioanalyzer (Agilent, Santa Clara, CA, USA), and ultimately sequenced on Illumina NextSeq 500 sequencer with 2 × 150 bp read length.

### 2.6. Short-Read Data Processing and Variant Detection

Short-read data processing and detection of somatic single nucleotide variants and copy-number variants was carried out as previously described [50]. In addition to the gatk-somatic-cnvs pipeline from GATK 4.1.4.0 [51], CNV calling was performed with CNVkit 0.9.10 [52] or CONTRA 2.0.8 [53] using standard parameters.

### 2.7. Long-Read Whole-Genome Sequencing and Data Processing

From the CD3+ sample, 100 ng DNA was subjected to library construction using the SQK-PBK004 kit and sequenced on the GridION sequencer with R9.4.1 flow cells (Oxford Nanopore Technologies, Oxford, UK). Base calling was performed with guppy 5.1.13 and the super-accuracy model. Reads were trimmed with Porechop 0.2.4 [54] and mapped to GRCh38 with minimap2 2.17 [55] using the -R, -Y, and the –MD parameters. Coverage analysis was performed by counting bases in 50 kb intervals using samtools and visualizing the results using a custom script. Structural variants were called using Sniffles 2.0.5 [56,57] with the –non-germline, –output-rnames, and –tandem-repeats parameters. The called SVs were filtered using a custom script available on GitHub (https://github.com/carstenhain/sv-analysis, accessed on 6 August 2022). SVs were visualized using Circos 0.69 [58] and custom python scripts.

### 2.8. Long-Read Targeted Sequencing

A total of 1 µg of DNA from the unprocessed spindle biopsy was subjected to library construction using the SQK-LSK109 kit and sequenced on the GridION sequencer with R9.4.1 flow cells (Oxford Nanopore Technologies, Oxford, UK). The adaptive sampling option was enabled in MinKNOW 20.10.6. For targeting, a bed-file and the human genome GRCh38 were used. The intervals were constructed manually, consisting of individual SV breakpoints with 15 kb padding as well as larger regions, e.g., the complete chr9p region. In total, 112 regions comprising 112 Mb were targeted with this approach (Appendix A). Base calling was done with guppy 5.1.13 and the super-accuracy model. Reads were trimmed with Porechop 0.2.4 [54] and mapped to GRCh38 with minimap2 2.17 [55] using the -R, -Y, and the –MD parameters. SNP calling was performed with longshot 0.4.1 [59], SNPs were filtered for QUAL > 100, and reads were phased with WhatsHap [60].

### 2.9. Detection of RAG or Microhomology Sites at SV Breakpoints

Analysis of RAG heptamers was carried out as described [20]. For microhomology detection, the sequences 12 bp around SV breakpoints were extracted and pairwise aligned using a custom script. The best alignment (in bp length) was used for further analysis. For comparison with RAG and microhomology occurrence in a random background, a random set of 100 SVs was generated and subjected to the same analysis steps as the actual SVs.

### 2.10. Detection of Templated Insertions

During SV filtering, all supporting reads from single SVs were assembled using lamassemble [61]. SV assemblies were mapped to GRCh38 with minimap2, and assemblies mapping to more than two regions in the genome were flagged and inspected manually.

### 2.11. Validation of SV Breakpoint Sequences Using Sanger Sequencing

Selected SVs detected by nanopore whole-genome sequencing were validated by Sanger sequencing. PCR products for sequencing were prepared using individual primers (Appendix A), cleaned up with ExoSAP-IT™ PCR Product Cleanup Reagent (ThermoFisher, Waltham, MA, USA), and sequenced using one PCR primer. Results were mapped and analyzed with Geneious 2020 (Geneious, Auckland, New Zealand).

### 2.12. Assembly and Identification of T-Cell Receptor Gamma Alleles

Reads spanning the two most common V/J combinations of the T-cell receptor gamma (*TRG*) were extracted and assembled using lamassemble [61]. The assembly was annotated with MiXCR [62]. For further analysis, reads were mapped to the assemblies and reads with multiple errors in the complementary determining region (CDR3) were removed. The assemblies were mapped to GRCh38 using minimap2 2.17 [55].

## 3. Results and Discussion

### 3.1. Early-Stage Mycosis Fungoides Biopsies Show Low Tumor Purity

We sought to quantify the fraction of tumor cells in sequenced samples from different forms and stages of CTCL. For this analysis, we gathered CTCL whole-exome (WXS) or whole-genome sequencing data. After quality control, 72 SS [20,23,25] and 4 tumor stage (T) 1 MF (this work) as well as 16 tumor stage 3 MF [22,23] (and this work) remained. For each sample, somatic SNVs and CNVs were called and tumor cell fraction was calculated with ABSOLUTE [63] (Figure 1). As expected, the T1 MF samples showed the lowest tumor cell fraction (median: 0.26; range: 0.24–0.28) followed by the T3 MF samples (median: 0.4; range: 0.26–0.69), while the SS samples contained the highest fraction of tumor cells (median: 0.9; range: 0.28–1.0). Individual outliers, such as T3 MF samples with more than 60% tumor cells, were observed. Although the gathered dataset is small, especially for early-stage MF, we believe that our results indicate a general trend: that tumor fraction in early-stage MF is lower than in later stages. This is coherent with previous studies, showing that malignant cells in MF lesions are in the minority [32] in early stages and the proportion of malignant cells increases in established lesions [34]. However, the trend is strongly influenced by other effects such as patient-specific variance of the tumor fraction or sampling bias (e.g., biopsy size and position relative to the patch). In general, low tumor fraction in early-stage MF poses a major obstacle for high-confidence analysis of somatic mutations in this sample group. To overcome this issue, a bias-free enrichment of tumor cells from biopsies before molecular analysis is necessary.

### 3.2. Tumor Cell Enrichment Leads to Increased Sensitivity for Copy-Number Variation Detection

We aimed to analyze one early-stage MF sample with special interest to high-confidence and -sensitivity CNV calling. Furthermore, we proposed that enrichment of T cells via CD3 or CD4 and the associated depletion of the surrounding cell types, such as keratinocytes, increases the tumor content of the sample.

Eight fresh 5 mm punch biopsies of an MF patient with proven T-cell clonality in stage IB (Appendix A) were randomly split into two batches before dissecting each batch into single cells. From the batch designated for CD3+ enrichment, one aliquot of cells was kept as the unprocessed sample. CD3+ cells were isolated from the rest using CD3 MicroBeads. From the other batch, CD4+ cells were isolated using CD4 MicroBeads. A punch biopsy of uninvolved skin was used as a healthy control. From all four samples, DNA was isolated and subjected to whole-exome sequencing. The unprocessed and enriched tumor samples were sequenced to a depth of 192×, 170×, and, 149×, respectively. Somatic variants, SNVs and CNVs, were called and absolute copy-numbers as well as tumor fraction were analyzed using ABSOLUTE [63].

The fraction of tumor cells in the unprocessed biopsy was 0.24, 0.58 in the CD3+, and 0.45 in the CD4+ enriched sample. These results depict an enrichment of tumor cells by the factor of 2.42 in the CD3+ and by 1.88 in the CD4+ sample. The increased tumor fraction in the CD3+ samples compared to the CD4+ sample is unexpected as MF samples, including this patient, are mostly CD3+CD4+CD8- [64]. We assume that the difference lies in the experimental procedure, especially a less rigorous tissue disruption in the CD4 batch due to the sensitivity of CD4 towards one of the normally used, but in this case omitted, protease. Further experimental optimization might resolve this problem, leading to a more complete tissue lysis and efficient capturing of CD4+ cells. Additionally, imbalances in the tumor fraction between both starting batches are possible. The seemingly more robust purification of malignant MF cells via CD3 should work well for most cases as the ratio of CD4:CD8 cell in MF is consistently above 1 [65,66], thus, the larger fraction of CD3+ cells should be CD4+ as well. Furthermore, the purification via CD3 has the additional advantage in capturing the malignant cells in rare CD4-CD8- MF cases, which are mostly CD3+ [67]. In the future, a more sophisticated antigen combination and FACS purification might lead to an even higher enrichment of malignant cells from individual biopsies. However, careful consideration of the antigens presented by the malignant cells is vital for successful enrichment. As MF cells are known to lose pan T-cell antigens [64], careful immunohistochemistry is necessary prior to enrichment to detect (subclonal) loss of the enrichment antigen. In the case of subclonal antigen loss, only a subpopulation of malignant cells would be enriched, and therefore the enriched cells would not represent the entirety of malignant cells in the tumor. Besides immunohistochemistry, T-cell receptor (TCR) sequencing is also crucial to elucidate the clonal architecture of the lesion. In MF, there are cases with monoclonal TCR but also cases with oligoclonal TCR, the latter strongly indicating multiple subclonal populations in the lesion [28]. Additionally, TCR sequencing can be used to track complete enrichment of all subpopulations of malignant cells by sequencing prior and after enrichment and comparing the results.

Analysis of the somatic mutations revealed 762 SNVs in the CD3+ enriched sample. Further classification of SNVs with mutationTimeR [68] resulted in 737 clonal and 25 subclonal SNVs, corresponding to a clonal mutation rate of 14.5 mutations per Mb. Over 85% of clonal SNVs were C>T transitions. Of the 737 clonal SNVs in the CD3+ sample, 736 (99.8%) were found in the CD4+ and 716 (97.2%) were found in the unprocessed sample. Known oncogenic mutations such as *MAPK1* p.E322K and *RHOA* p.A161P were detected in all samples, albeit with 3.37× or 2.76× higher frequency in the CD3+ sample than in the unprocessed sample, respectively (Figure 2a). The gain-of-function *RHOA* p.A161P mutation [69] is similar to the mutation occurrence in other CTCL cases outside of the *RHOA* p.G17 hotspot commonly mutated in other cancers and lymphoma [70]. The gain-of-function *MAPK1* p.E322K mutation [71] is found exclusively in MF and not in SS [70].

In the CD3+ sample, 7.16% of the genome is hetero- and 0.03% homozygously deleted, while 10.28% of the genome is amplified. A total of 69 amplified or deleted segments were called. Similar results (60 amplified or deleted segments, visual agreement between both samples) were observed for the CD4+ sample (Appendix A). In the unprocessed sample, CNV calling classified 8.7% of the genome as deleted and 10.08% as amplified. Only 14 amplified or deleted segments were found. A clear distinction between homo- and heterozygous deletions was also not possible. CNV analysis using other CNV callers such as CONTRA [53] or CNVkit [52] yielded more noisy results (Appendix A).

In general, the unprocessed sample showed a much less sensitive CNV calling than the CD3+ sample. Exemplary and of special interest is the CNV on chr9 (Figure 2b). CNV calling in the CD3+ sample clearly shows two different levels of deletions: a large single-deleted region from 2.8 to 22.4 Mb and a small double-deleted region from 21.8 to 22 Mb. The tumor suppressor gene *CDKN2A*, which is located in the double-deleted segment on chr9:21.9 Mb, is thus homozygously deleted in this patient. CNV calling in the unprocessed sample cannot resolve these deletions and proposes one large weakly deleted region from chr9:0-64 Mb.

Insensitive CNV calling, especially for chromosomal segments considerably smaller than a chromosomal arm, is observed in the unprocessed sample at numerous locations. This includes failure to detect *STAT3/5* amplification and inaccurate detection of an *ARID1A* deletion (Appendix A).

A detailed analysis of this study’s patients showed multiple mutations recurrent for CTCL and partly specific for MF, including the *RHOA* and *MAPK1* SNVs as well as *CDKN2A* deletion, *STAT3/5* amplification, *ARID1A* deletion and chr7 trisomy [19,20,70,72]. The overlap in called SNVs between the unprocessed and the tumor enriched samples was very high, and especially the oncogenic mutations were called confidently in all samples.

This result is expected as WXS, without additional modifications, has a limit of detection at variant allele frequencies of 5–10% for SNVs [36,37,38]. In contrast, CNV calling in WXS is a difficult problem even in germline samples with presumable completely clonal CNVs [40]. CNV calling performs better in short-read WGS, but even there, little overlap between existing tools is reported [40,41], and an allele fraction of 20% is considered the minimum for accurate SV calling [43].

In tumor samples, this problem is aggravated due to lower fractions of tumor cells containing the CNVs, resulting in weaker signals of individual CNVs in the WXS data. In this concrete example, the signal-to-noise ratio of the unprocessed MF biopsy of single allele amplifications or deletions are close to 1 (1.12 and 1.07, respectively), while the signal-to-noise ratio of the CD3+ sample is 2.5 (2.6 and 2.55, respectively) (Appendix A). This difference in signal-to-noise ratio is the major reason for the insensitive CNV calling in the unprocessed sample. For large CNVs such as the trisomy 7 or the general mention that chr9p is affected by some kind of deletion, this dataset is sufficient. However, for a more detailed analysis of the individual CNVs, a more precise resolution of the CNV breakpoints and a better classification of how many alleles are deleted or amplified is crucial.

This can be illustrated using two examples:

The *STAT3/5* gain on chr17q is the most recurrent amplification in CTCL and is present in 60% of SS cases [70]. In SS [24,26,73] and myeloid neoplasms [74], this CNV is often caused by a chr17 isochromosome, leading to a simultaneous loss of 17p including *TP53*. In this sample, the *STAT3/5* genes are presumably amplified as a 466 kb segment, separate from another CNV, leading to amplification of the distal part of chr17q (starting from 43.5 Mb). This result indicates that the *STAT3/5* amplification in this sample is not due to an isochromosome or a large-scale amplification, but rather due to a smaller amplification. This might indicate another mutation mechanism than the one leading to isochromosomes [75].

Secondly, the homozygous *CDKN2A* deletion is detected in high resolution only in the CD3+ sample. In the unprocessed sample, the distinction between hetero- and homozygously deleted segments is not possible. The distinction between homo- and heterozygous deletion is important for the correct interpretation/prediction of the effect of the mutation. In case of haploinsufficient genes such as *CDKN2A* [76], loss of one allele already leads to a phenotypic change. Homozygous deletions lead to lower mRNA levels than heterozygous deletions for *CDKN2A* [77,78], indicating an even stronger effect. In *CDKN2A* and other tumor suppressors, a homozygous deletion—or an otherwise homozygous loss by a combination of deletion, SNV, or promoter methylation—leads to more severe disease with decreased overall survival [79,80,81]. However, in certain cases, stark differences in phenotype between hetero- and homozygously deleted/mutated genes are observed [82].

Taken together, only sufficiently large fractions of tumor cells enable analysis of the correct location and state of individual CNVs. This is an extremely important point for further research in MF as a whole, as CTCL does not show highly recurrent SNVs but strong recurrence in CNVs, highlighting the great importance of CNVs in CTCL [20]. This fits with our previous analysis suggesting that CNVs are among the earliest mutations in the pathogenesis of SS [50]. For the ongoing endeavor to build genomic knowledge of the currently underrepresented early-stage MF and potentially discern progressing from indolent forms, the following points have to be considered. In early MF pathogenesis, CNVs might also be the main unifying mutations, and low tumor fraction, which is present in early MF lesions, will hinder accurate and confident CNV analysis, thus hiding the main genetic change in MF. Therefore, careful and preferably bias-free enrichment of tumor cells above a certain threshold is mandatory to achieve high-quality data needed to build a complete picture of genetic changes in early MF. In this context, it is important to consider that enrichment of tumor cells does not capture the original (genetic) composition of the tissue, including the microenvironment. It makes a compromise to allow the detection of genetic changes in the tumor cells with increased confidence, on the expense of missing surrounding mutations. If changes in the tumor microenvironment are of special interest, other techniques such as single-cell RNA sequencing or spatial transcriptomics are better suited.

### 3.3. Nanopore Sequencing Reveals the Structural Variants Underlying Classical MF CNVs

WXS detects amplifications or deletions but does not locate the breakpoints of the causative structural variants (SVs) leading to these CNVs. As such, the exact class of SVs and possibly the biological mechanism leading to these SVs is not resolvable with WXS. For SV analysis, third generation long-read sequencing shows increased performance, especially in repeating or low-complexity regions [45], and enables direct identification of complex SVs consisting of multiple translocations.

To elucidate the SVs leading to the observed CNVs, we used nanopore long-read genome sequencing on the CD3+ sample. Specifically, we utilized the PCR barcoding kit and generated 31.7 million reads with a mean length of 2.1 kb, resulting in a 22.2× mean genome coverage. The application of a PCR kit was a compromise due to low sample DNA quantities. Sequencing of native long fragments enables more confident SV calling [56] and simultaneous analysis of the epigenome [83].

Structural variants were called with Sniffles2 [56,57] and filtered using a custom script. In the following, only SVs longer than 5000 bp or inter-chromosomal translocations are considered, thereby removing three quarter of germline SVs (Appendix A). Obviously, a correct tumor-normal-approach for removal of germline SV would be preferable and would enable correct identification of smaller somatic SVs. Under these criteria, 167 SVs (93 deletions, 16 inversions, 10 duplications, and 48 inter-chromosomal translocations) were found.

Combined plotting of SVs and CNVs yields an insight into the genomic architecture of the sample (Figure 3). In most cases, a breakpoint is located at (or close to) the edge between two genomic segments of different copy-number. These breakpoints are often clustered, as exemplified by the dense clusters on chr1, chr9, and chr17.

Special attention was directed to the homozygous deletion of the tumor suppressor gene *CDKN2A* already detected in the WXS data (Figure 2b). Genome sequencing revealed three large SVs: one 512 kb deletion and two inversions (10.7 Mb and 31.6 Mb) (Figure 4a). All SVs were validated by Sanger sequencing (Appendix A). The deletion fits to the homozygously deleted region affecting *CDKN2A*. The two inversions are unbalanced, meaning that only one side of the original DNA break is rescued (Appendix A). Both inversions are *in cis* as shown by phasing of targeted long-read sequencing data (Appendix A). This leads to a complex SV consisting of a deletion from 2.8 to 34.5 Mb, visible in the coverage data (Figure 2b), coupled with an inversed insertion of chr9:23.8–34.4 Mb.

Another important CNV found in the WXS data was an amplification on chr17 containing the *STAT3/5* genes. Translocations to chr16 and chr9 are found at the left and right border of the amplified segment, respectively (Figure 4c). If both SVs were *in cis*, the *STAT3/5* amplification would be explained by an SV of the chain of templated insertion class [84]. This would lead to a fusion of chr9 to chr16 with the 495 kb fragment of chr17, containing the CDS for *STAT3/5* in-between. However, *in cis* localization cannot be proven since a long stretch of homozygosity prevents successful phasing here (Appendix A).

A further look into the detected SVs revealed more validated or assumed SVs of the templated insertion class. One validated example of another bridge of templated insertion, with single reads spanning the complete insertion and both adjacent breakpoints, is a 2.2 Mb deletion on chr1 in which a 2.8 kb fragment originally located at chr1:28.2 Mb is inserted (Appendix A). The location of the inserted fragment is extremely close (639 bp) to a breakpoint of another 4.8 Mb deletion, which leads to heterozygous deletion of *ARID1A* (Figure 4b).

In addition to these SVs of the templated insertion class (one assumed and one validated), seven more templated insertion translocations were spanned completely by individual reads (Appendix A). The insertion length was between 219 bp and 4,036 bp. Six of these SVs carried one insertion and one SV had two insertions.

Besides these complicated but (putatively) resolvable SVs, chr5q shows a much more complex rearrangement (Figure 4d). In a 53 Mb region, 15 intrachromosomal and 11 inter-chromosomal translocations to chromosomes 1, 3, 9, and 17 were identified. The high density of breakpoints leading to the same and other chromosomes indicates a chromoplexy-like event for chr5.

Our analysis resolved the causative SVs for some of the most recurrent CNVs in MF, such as the *STAT3/5* gain and *ARID1A* and *CDKN2A* loss. Furthermore, it is exemplary for reconstructing the genomic structure in regions with multiple SVs by combining information on called SVs, copy-number, and (where possible) phasing. We believe this approach to be very important for the CNV-rich disease MF: accurate determination of individual SVs and allelic resolution of SVs (and SNVs) will provide the basis for a deeper understanding of the genomic changes during MF development.

Another important aspect shown in this work is the direct detection on templated insertion SVs in this sample. This class features insertion of one or more genomic segments into a simple translocation, thus leading to its amplification. For the insertion size of this SV, different data are given, ranging from a few bp [85] to 50–100 bp [86], and several Mb [84]. Using short-read sequencing, long templated insertions can only be detected indirectly (e.g., by comparison of allele frequencies [84]). Long-read sequencing has the potential to directly span the complete insertion with one read and validate the templated insertion this way. Thereby, longer reads increase the size of directly detectable templated insertion SVs. Alternatively, phasing of the complete insertion segment or both SV breakpoints into one haplotype enables confident determination of a templated insertion SV. Templated insertion SVs are an erroneous repair product of double-strand breaks [85,86]. As potential mechanisms, the polymerase theta-mediated end-joining pathway [85] or insertion of RNA-derived sequences [86] are discussed. If other MF patients also show multiple templated insertion SVs, this SV class and its associated mechanism could be part of the processes leading to the extensive SVs in MF.

Besides the determination of templated insertions, further detection of common mechanisms for SV formation was unsuccessful. In other lymphoid malignancies, such as chronic myeloid leukemia [87] or T-cell acute lymphatic leukemia [88], aberrant recombination-activating gene (RAG) activity and associated breakpoints close to cryptic RAG recognition sites are observed. This enrichment was discussed in CTCL [20], but no significant enrichment of cryptic RAG recognition sites are detected in this sample (Appendix A). While some SVs show signs of microhomology [89,90] (Appendix A) as a whole, no significant enrichment of microhomology is seen compared to a random background (Appendix A). The question of the mechanism behind SV formation in MF remains open, as does the question of whether it is one predominant mechanism or a mixture of different pathways. To tackle this question, a larger cohort of WGS with accurately called SVs seems necessary. Further information, e.g., regarding the 3D structure [91] or the temporal sequence of individual mutation processes [68], might have to be integrated to obtain a complete picture. In addition to the large SVs, we identified TCR rearrangements for the gamma chain. We assembled the two predominant alleles of this sample (TRGV10-TRGJ1 and TRGV8-TRGJP1) (Appendix A). Unfortunately, the reads from this library are too short for both phasing and assembly of the complete TCR gamma region or the more complex TCR beta region (data not shown). Using native long reads, such an assembly is possible and revealed large SVs, including insertions of additional V-segments [92]. As such, a more extensive analysis of germline structure of MF’s patients TCRs would be feasible and of interest.

## 4. Conclusions

Genomic data for MF account for only a subproportional share of the complete knowledge of genomic changes in CTCL, in particular for early-stage MF. One major obstacle in elucidating the genetic changes in this underrepresented part of CTCL is the low tumor fraction in early-stage MF, which makes the detection of somatic variants and especially CNVs difficult. In this work, we outlined a workflow consisting of dissociation of the skin biopsy and subsequent antigen-based cell capture, thereby increasing tumor fraction and enabling high-sensitivity CNV calling. Using this workflow with CD3+ based enrichment for an early-stage MF sample, the tumor fraction more than doubled. This improvement allowed high-quality calling of the extremely complex CNV landscape of this sample, which was not visible without enrichment. Specific examples only detectable in the enriched sample are the homozygous deletion of *CDKN2A* and focal amplification of *STAT3/5*. The sample preparation presented in this study makes MF samples from early stages more accessible for genetic diagnostics. In future work, this enables the exploration of genetic differences, including the highly frequent and recurrent CNVs, between indolent and progressive patients already in early stages of MF. In addition, we applied long-read nanopore sequencing as a tool for detection of complex SVs. Our results elucidated the SVs leading to recurrent CNVs in MF, namely *ARID1A* deletion and *STAT3/5* amplification. Furthermore, long reads allow phasing of multiple SVs into complex haplotypes. Thereby, we succeeded in resolving the deletion of *CDKN2A* as well as identifying multiple templated insertion SVs. Our work introduces an efficacious methodology for unraveling the genetic changes in low tumor fraction MF samples, either at CNV level by WXS or by complete resolution of SVs using long-read WGS. Due to the simplicity of the technical set-up and the importance of CNVs and SVs in MF, our method is directly applicable in a large cohort study of early-stage MF patients to expand knowledge on the genetic landscape of MF.

## Figures and Tables

**Figure 1 cancers-14-04466-f001:**
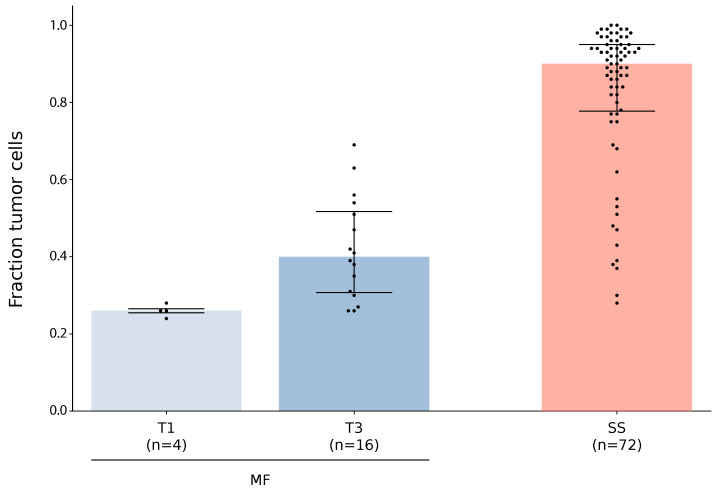
Fraction of tumor cells in samples from different forms and stages of CTCL. WXS or WGS data from this work or from published studies [20,22,23,25] were evaluated for tumor cell fraction using ABSOLUTE [63]. The samples are grouped and classified, according to patient information, into MF with tumor stage 1 or 3, or SS. For each class, sample size, median, and first and third quantile are depicted by horizontal bars.

**Figure 2 cancers-14-04466-f002:**
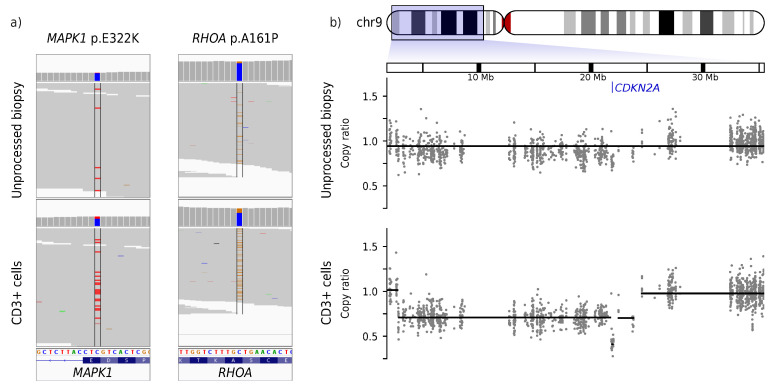
Somatic variant calling in an early-stage MF skin biopsy without (top) and with (bottom) enrichment of tumor cells. One skin lesion of a patient with tumor stage 1 MF was biopsied. One half (unprocessed biopsy) was used without further steps. The other half of the material was used for enrichment of CD3-positive cells (CD3+ cells). Afterwards, both halves were used for DNA isolation, WXS, and calling of somatic SNVs and CNVs. (**a**) Raw data for the somatic SNVs *MAPK1* p.E322K and *RHOA* p.A161P where colored strips indicate variant reads diverging from the genome reference. (**b**) Raw and processed data for a deletion on chr9 leading to homozygous loss of *CDKN2A*. The gray dots show the copy-ratio (denoised and normalized read depth of individual exons) while the black lines depict the automatic CNV call. A copy-ratio of 1 indicates a neutral copy-number of 2 on autosomes. The genomic position of *CDKN2A* is marked in blue.

**Figure 3 cancers-14-04466-f003:**
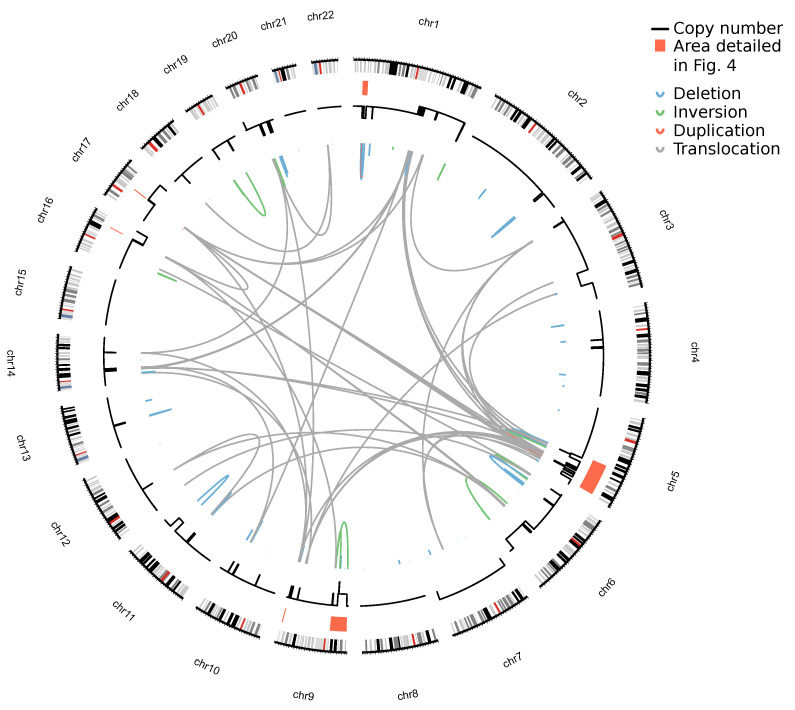
Structural variations in a early-stage MF sample detected by nanopore sequencing. Plotting of the copy-number landscape as derived from the WXS (black lines, outer ring) and the SVs detected by nanopore sequencing (colored lines, inner ring). Coloring of individual SVs indicates the SV class. The height of SV curves is proportional to the SV size. The areas marked in red are elucidated in Figure 4.

**Figure 4 cancers-14-04466-f004:**
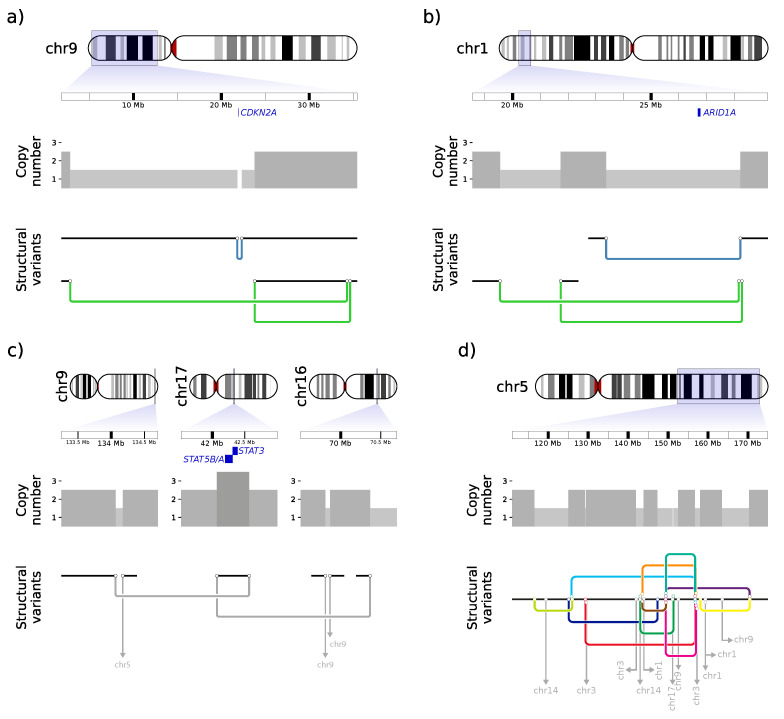
Deletion of *CDKN2A* and *ARID1A*, amplification of *STAT5* by templated insertion SVs and complex rearrangement of chromosome 5. Each panel consists of three parts (top to bottom): The genome position of the respective section with marked position of the gene of interest (blue), the copy-number, and the detected SVs. SVs consist of two breakpoints (black circles) connected by a junction, colored dependent on the type of SV (deletion: blue, inversion: green, translocation: gray). Translocations where only one breakpoint is visible are shown as gray arrows and the chromosomal fusion partner is annotated. Chromosomal segments joined by SVs are depicted by black lines. (**a**) Homozygous deletion of *CDKN2A* by two SVs *in cis*: one small deletion and one bridge of templated insertion SV, where chr9: 23.8–34.4 Mb is inserted inversely in a chr9: 2.8–34.5 Mb deletion. (**b**) Heterozygous deletion of *ARID1A* and an adjacent bridge of templated insertion SV with an insertion of a 2.8 kb segment only 639 bp distant to one *ARID1A* deletion breakpoint. For this SV, single reads spanning both breakpoints as well as the complete inserted region are found. (**c**) Amplification of *STAT3/5* by a chain of templated insertion SV between chr9, chr17, and chr16. (**d**) Complex rearrangement on chr5 with multiple intra- and inter-chromosomal translocations. For a better overview, the SVs are depicted in different colors and SVs with templated insertions are marked by additional breakpoint symbols. Due to the complexity of the region, reconstruction into a potential model was not attempted.

## Data Availability

For tumor cell fraction in MF and SS, the dbGaP datasets phs000913, phs000725 and the SRA datasets SRP058948 and SRP059214 were used with appropriate permissions. The data generated during this study are available on request from the corresponding author.

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
