# Peer review of "Unraveling the Structural Variations of Early-Stage Mycosis Fungoides—CD3 Based Purification and Third Generation Sequencing as Novel Tools for the Genomic Landscape in CTCL"

_cancers, 2022, doi:10.3390/cancers14184466_

Round 1
Reviewer 1 Report
Identifying copy number variations and mutational drivers in early-stage mycosis fungoides can be hindered by the inherent density of malignant T cells that reside in the skin. This manuscript by Hain et al. proposes that these limitations can be overcome by first enriching for CD3+ or CD4+ T cells from lesional skin. Using this approach, somatic variants (e.g., in MAPK1, RHOA) and deletions can be identified from skin biopsies of an early MF patients that would otherwise be undetected without pre-enrichment. The ability to uncover somatic variants by first pre-enriching for CD3 or CD4 T cells in lesional skin could be easily incorporated by others, and therefore may be of general interest to CTCL field
Minor concerns:
1. It seems like enrichment and somatic variant analysis were performed on both purified CD3+ as well as CD4+ cells. Data for CD3+ cells is shown in the main figures, while CD4+ data is shown in the supplemental. Did the somatic variants change depending on whether CD3 or CD4 cells were purified? Could a statement contrasting findings for the two approaches, or a figure comparing the findings be added?
2. Is there concern that by only performing somatic variant analysis on T cells that potential mutational drivers in non-T cell or stromal cells (e.g., keratinocytes, dendritic cells) would be missed?
Author Response
Please see attached document including our dispositions on the referees' suggestions and comments.

Reviewer 2 Report
In the current study, the authors analyzed the genomic landscape of early-stage mycosis fungoides (MF) using several types of samples. The authors found that genomic alterations could be more frequently identified in the DNA extracted from CD3-positive cells than in unfractionated cells. Also, the authors utilized long-read sequencing to identify structural variants using the CD3+ samples. Overall, the experiments are well performed, and the manuscript is well written.
1. It is unclear about the clonality evaluated from TCR analysis in the case that the authors analyzed. Did the tumor show monoclonal, or oligo-clonal from the standpoint of TCR rearrangement? Since it seems that only a part of malignant clones possesses the alterations, it is unclear whether the analysis using the CD3+ samples always gives good results. Ideally, the authors need to quantify and compare the TCR patterns among the CD3+, CD4+ and unfractionated cells to conclude the results.
Author Response

(The authors gave the same response as above.)

Round 2
Reviewer 2 Report
The authors have responded to the reviewer's comments and modified the manuscript. The changes have made the manuscript clearer.